# Clinical characteristics and outcomes of patients admitted with COVID-19 at a public-sector hospital over the first two waves of SARS-CoV-2 infection in Harare, Zimbabwe: A prospective cohort study

**Arun Fryatt**[1]*, **Trevor Chivandire**[2], **Victoria Simms**[1,3], **Perseverance Chikide**[2], **Trymore Munorwa**[2], **Ellane Simon**[2], **Lovemore Nyasha Sigwadhi**[1], **Katharina Kranzer**[1,4,5], **Tsitsi M. Magure**[2], **Aspect Maunganidze**[2], **Leolin Katsidzira**[2,6], **Rashida A. Ferrand**[1,4]

1 Biomedical Research and Training Institute, Harare, Zimbabwe, 2 Parirenyatwa Group of Hospitals, Harare, Zimbabwe, 3 MRC International Statistics and Epidemiology Group, Department of Infectious Disease Epidemiology, London School of Hygiene & Tropical Medicine, London, United Kingdom, 4 Clinical Research Department, London School of Hygiene & Tropical Medicine, London, United Kingdom, 5 Division of Infectious Diseases and Tropical Medicine, University Hospital, LMU Munich, Munich, Germany, 6 Internal Medicine Unit, University of Zimbabwe, Harare, Zimbabwe

* arunfryatt@gmail.com

**Data Availability Statement:** As informed consent was not obtained from participants, public

## Abstract

### Background

We investigated the clinical profile, complications, and outcomes of inpatients with COVID-19 at Parirenyatwa Hospital, Harare, across the first two waves of SARS-CoV-2 infection, and factors associated with mortality.

### Methods

We conducted a prospective cohort study on all patients admitted to the COVID-19 unit. Data were extracted from medical records and negative binomial regression with robust standard errors was used to assess the association between sociodemographic and clinical characteristics and mortality. Cox Regression was used for sensitivity analysis.

### Results

Of 563 people admitted with COVID-19 between 2 July 2020 and 19 March 2021, 214 (38.0%) died, 340 were discharged and 9 transferred. The median age was 56 (IQR 44–68) years and 53.8% were male. Overall, 38.8% experienced a complication, the most common being acute kidney injury (17.9%) and hyperglycaemia (13.1%). The most common comorbidity was hypertension (41.3%) followed by diabetes (28.6%), HIV (12.1%), cardiovascular disease (10.9%) and chronic kidney disease (7.8%). Among participants who stayed in the ward for more than 1 night, mortality was higher in patients with comorbidity compared to those without any comorbidity (38.7% vs 25.5%, risk ratio (RR) = 1.52 (95% CI 1.11, 2.07), p = 0.008). After adjusting for oxygen saturation, comorbidities, sex and pregnancy,

deposition of the dataset would breach Medical Research Council of Zimbabwe (MRCZ) requirements for data from a public hospital. Interested parties should contact the Parirenyatwa Group of Hospitals in Zimbabwe regarding access to the data. Institutional point of contact for all data queries: Mr Philip Mapurisa mutsamaps@gmail. com Lead Health Information Officer for Covid-19 Parirenyatwa Group of Hospitals Mazowe Street Harare Zimbabwe.

**Funding:** This work was supported by the ELMA Foundation and the Wellcome Trust (through a Senior Fellowship to RAF [Grant number 206316_Z_17_Z]). The funders had no role in study design, data collection and analysis, decision to publish, or preparation of the manuscript.

**Competing interests:** The authors have declared that no competing interests exist.

mortality was higher in the second wave than in the first (adjusted RR 1.23, 95% CI 1.00–1.51, p = 0.05). In the second wave 57/161 (35.4%) deaths were attributed to lack of resources, mainly human resources.

## Conclusion

The mortality rate was high and clinical COVID-19 care needs to pay careful attention to patient monitoring for complications and management of comorbidities. This will require addressing the critical health workforce shortage issues. Prevention of COVID-19 including vaccination particularly among individuals with comorbidities remains a high priority.

## Introduction

In late 2019 a cluster of pneumonias in Wuhan, China was found to be caused by the Severe Acute Respiratory Syndrome Coronavirus 2 (SARS-CoV-2) [1]. The disease caused by this virus was termed COVID-19 and the World Health Organization declared the SARS-CoV-2 outbreak a global pandemic on the 11th of March 2020. As of the 7th of January, 2022, total cases across the world surpassed 298 million of which over 10 million were on the African continent [2].

The first case of COVID-19 was reported in Zimbabwe on the 20th of March 2020 in a resident returning from the United Kingdom, and by the 7th of January 2022, 221,282 cases had been reported [2]. The true number of SAR-CoV-2 infections is probably much higher given that many SARS-CoV-2 infections are asymptomatic and testing capacity and population surveillance are limited [3–5]. The first three waves peaked in August 2020, January 2021, and July 2021, with most cases occurring in the two largest cities, Harare and Bulawayo [2]. A recent study reported a 53% SARS-CoV-2 seroprevalence in three high density suburbs in Harare following the second wave in April 2021 [6]. The second wave coincided with the emergence of a new SARS-CoV-2 variant B.1.351 (beta variant) first identified in South Africa which quickly became the dominant strain accounting for 95% of sequenced samples in January 2021 in Zimbabwe [3, 4].

Detailed clinical data of patients with COVID-19 and treatment outcomes in Africa are scarce. These data are vital for resources planning and to improve clinical management. A systematic review including 15 studies reported increased hospitalization and death in elderly people and patients with comorbidities. The review included four studies conducted in African countries and eleven multi-country studies [5].

The aim of this study was to investigate the clinical profiles, complications, and outcomes of patients with COVID-19 admitted at a public sector hospital in Harare during the first and second SARS-CoV-2 waves.

## Methods

### Study design and setting

We conducted a prospective cohort study extracting routinely recorded clinical data on all patients admitted to the COVID-19 unit at Parirenyatwa Group of Hospital (PGH) in Harare, Zimbabwe between the 17th of June 2020 and the 19th of March 2021. The time-period between 17th of June and 31st October 2020 was defined as first wave in Zimbabwe, the time-period between 1st of November 2020 and 31st of March 2021 was defined as second wave.

PGH is a public teaching hospital associated with the University of Zimbabwe Medical School and accepts referrals from hospitals across the country, providing tertiary level care. A

dedicated COVID-19 inpatient centre was established in June 2020 including 3 medical wards with a total of 90 beds with piped oxygen availability, as well as a critical care ward with four beds, with facilities for invasive ventilation.

There had been intermittent periods of industrial action by healthcare workers over remunerations, working conditions, and a critical shortage of medical equipment and supplies in the two years preceding the pandemic. Doctors and nurses had taken industrial action over working conditions including the lack of personal protective equipment (PPE) in 2020 disrupting clinical services including provision of haemodialysis. Staff partially returned to work in September 2020. Due to the industrial action during the first COVID-19 wave, volunteers were recruited by the hospital to provide clinical care and therefore patient to physician/nurse ratios were variable but significantly more limited than usual. Furthermore, critical care nursing staff and medical personnel to manage ventilation were not available in the first wave.

Treatment and management followed the national case management and infection prevention & control guidelines [7, 8]. Briefly, the mainstay of care was oxygen therapy, prophylactic anticoagulation, and intravenous fluids (where oral fluid intake was inadequate). Patients with severe disease defined as oxygen saturations of $< 94\%$ on room air were given a 10-day course of dexamethasone (6mg) [9]. A random point of care blood glucose was measured in all patients to screen for diabetes. Those with raised blood glucose ($\geq$7mmol/L) and/or symptoms suggestive of diabetes had glycated haemoglobin (HbA1c) measured. Patients were tested for HIV infection on admission unless previously diagnosed. Where indicated, patients received treatment with high flow nasal oxygenation (HFNO) and/or continuous positive airway pressure (CPAP) ventilatory support on the general ward. Antibiotic therapy was given when bacterial (super-) infections were suspected or could not be excluded. Complications were managed according to standard hospital protocols.

## Data collection

Anonymized data were extracted from the medical notes following discharge, transfer, or death. Data included age, sex, province of residence, clinical history, signs, symptoms and blood results at presentation, treatment, complications, and outcome. Patients without molecular confirmation of SARS-CoV-2 infection with signs and symptoms consistent with COVID-19 disease, based on the case definition in national guidelines, in the absence of a more likely diagnosis were categorized as "clinical COVID-19" [7]. The primary cause of admission was recorded for patients with molecular confirmed incidental SARS-CoV-2 infection without signs and symptoms of COVID-19 (incidental COVID-19 patients).

Newly diagnosed diabetes was based on the American Diabetes Association's guidance on HbA1c diagnostic cut-offs [10]. Previous comorbidities were ascertained through self-report or/and from patient-held medical records. Acute Kidney Injury (AKI) was defined using the United Kingdom's National Institute of Clinical Excellence standards [11]. A considerable number of patients were too unwell for measurement of height and weight. Therefore, weight categories were based on a visual assessment by the clinical team. From 1st November 2020, the treating clinician recorded whether care could and should have been escalated to prevent deterioration, but due to a lack of resources (e.g., lack of ICU (Intensive Care Unit) staff) this was not possible.

## Statistical analysis

Statistical analyses were performed using Stata V16 (StatCorp, College Station, Texas, USA). Categorical variables were expressed using frequencies and percentages. The dependent variable was mortality. The independent variables were age, sex, location, oxygen saturation,

weight category, haemoglobin (gram/decilitre), white blood cells ($10^9$/litre), creatinine (micromol/litre), any comorbidity, and pregnancy. Log-rank tests were used to assess the difference between time to death by wave and duration of admission. Due to overdispersion in the data, negative binomial regression with robust standard errors was used to assess the association between demographic and biochemical characteristics and mortality. Factors associated with death at p-value <0.15 in unadjusted univariable negative binomial regression were included in a multivariable model to identify variables associated with mortality in patients who stayed more than one night in the unit. Adjusted risk ratios and their 95% confidence intervals (CIs) were used as a measure of association. Factors with p<0.05 were considered significantly associated with mortality. A directed acyclic graph (DAG) was constructed to identify variables expected to be associated with both wave and mortality, and these variables were adjusted for. As a sensitivity analysis, Cox regression with robust standard errors was used to estimate the mortality hazard ratio in patients who stayed more than 1 night, adjusting for the same variables used in the negative binomial multivariate regression.

### Ethical approval

Ethical approval was obtained from the Medical Research Council of Zimbabwe (MRCZ/E/ 292) and Parirenyatwa Group of Hospitals. As data was anonymized, individual consent for participation was not obtained.

## Results

Between 2 July 2020 and 19 March 2021, 659 patients were admitted to the Unit, of whom 563 were diagnosed with COVID-19 and 96 tested SARS-CoV-2 positive but were admitted for other reasons (incidental COVID-19 patients). The number of daily admissions mirrored the national SARS-CoV-2 notifications (Fig 1), with the first wave peaking in early August 2020 and the second, larger wave peaking in early January 2021.

Of the 563 patients diagnosed with COVID-19, 515 reported that they had a SARS-CoV-2 polymerase chain reaction (PCR) performed: 365 had a positive result, 29 were negative, and 121 were unknown. The remaining 48 patients had a SARS-CoV-2 antigen test, of which 23 were positive and 25 were unknown.

Of the 96 incidental COVID-19 patients, 45 were admitted during in wave one and 51 during wave two. The reasons for admissions were pregnancy associated complications (n = 21), infection (n = 11) (e.g., typhoid fever, tuberculosis), chronic kidney disease (n = 10), surgery (n = 10), cardiovascular disease (n = 9) (e.g., heart failure), diabetes (n = 7), trauma (n = 5), severe anaemia (n = 5), neurological (n = 3) (e.g., seizures, stroke), 15 others (n = 15) (e.g., no transport home).

### Characteristics of COVID-19 patients

Descriptive characteristics of patients with COVID-19 are shown in Table 1 stratified by wave. The median age was 56 (IQR 44–68, range 3–100) years, 303 (53.8%) were men and 10% were transferred from outside Harare. The most common comorbidity was hypertension (41.3%) followed by diabetes (28.6%), HIV (12.1%), cardiovascular disease (10.9%) and chronic kidney disease (7.8%). Two thirds of patients (69.3%) had at least one comorbidity. At admission 41.3% had an elevated blood pressure, and 7.8% had an oxygen saturation <88% despite being on oxygen.

Age and sex of patients were comparable across waves. Patients admitted during the second wave were more likely to be residing outside of Harare (12.2% vs 4.5%), less likely to have a comorbidity (67.2% vs 75.3%), specifically diabetes and cardiovascular disease, more likely to

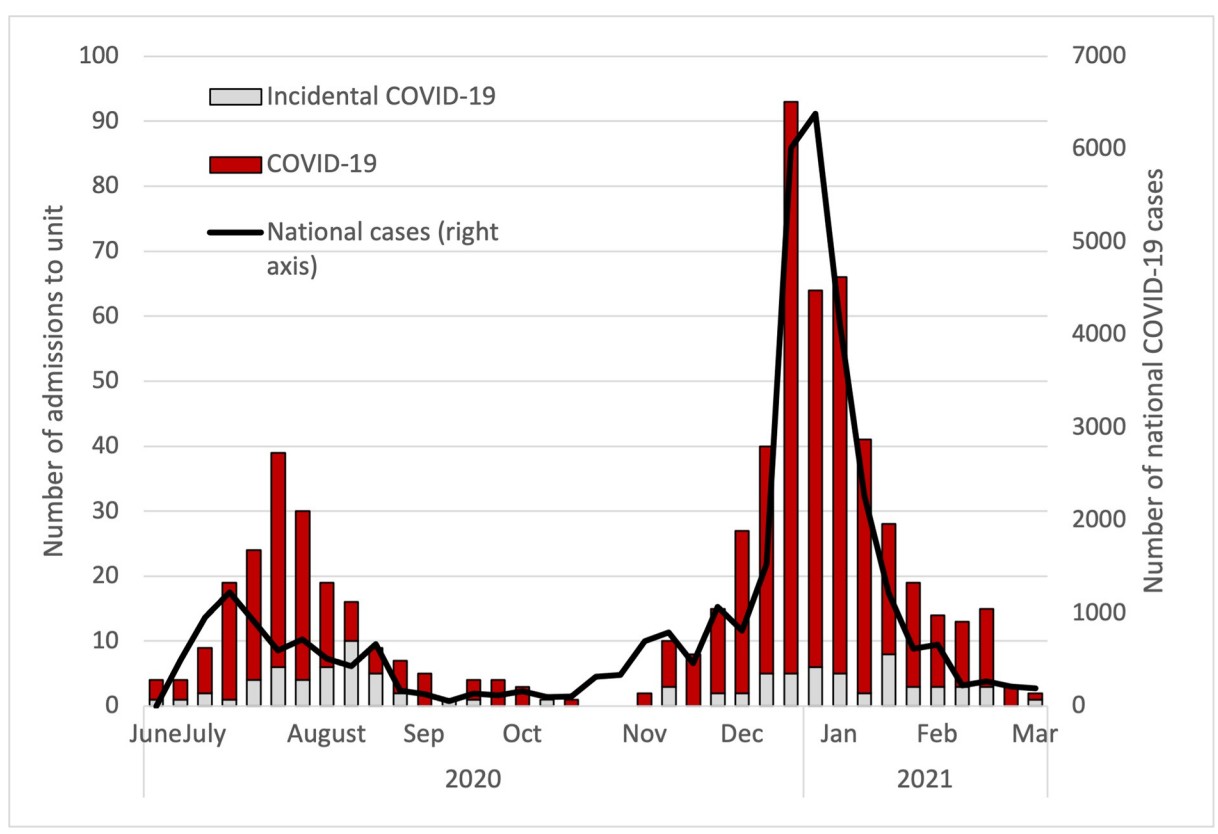

**Fig 1. COVID-19 and incidental COVID-19 admissions over time, and national COVID-19 case reports.**

be pregnant (4% vs 0.7%) and have a higher oxygen saturation at admission (saturating $\geq 88\%$ at admission, 32.5% vs 23.4%). A total of 215 patients (38.8%) experienced at least one complication with the most common complication being acute kidney injury (n = 99, 17.9%) followed by hyperglycaemia (n = 72, 13.1%).

Of the 554 patients, 50 participants (9.0%) received either HFNO or CPAP and 25 (4.5%) were referred to ICU. 226 patients (40.8%) received antibiotics, most frequently ceftriaxone as this was the antibiotic mentioned in the national COVID-19 treatment guidelines [7]. Compared to the first wave, a lower proportion of patients (6.9% vs 14.3%) in the second wave received HFNO and/or CPAP, and a higher proportion (44% vs 33.1%) received an antibiotic.

### Outcomes of COVID-19 patients & factors associated with mortality

Among 563 COVID-19 patients: 9 (1.6%), 340 (60.4%) and 214 (38.0%) were transferred out, discharged, and died respectively (see Table 2). Of the 25 patients in ICU 20 died (80.0%). Seven of the twenty-five ICU patients were intubated (one in wave 1 and six in wave 2) and four died (4 in wave 2). In wave 2 there was an increase in the proportion who were admitted to ICU and/or intubated for mechanical ventilation (5.1% vs 2.6%), although overall numbers were low. Mortality was comparable in the first (58/153, 37.9%) and second (156/401, 38.9%) wave (p = 0.29). The median time in hospital was 4 days (IQR 2–8, range 0–95) for those who died and 6 days (IQR 3–11, range 0–66) for those who were discharged. Of those patients who died, one in five (47/214; 22.0%) died within 24h of admission (Fig 2). Mortality was 28.2%, 36.2% and 55.8% among patients with no, one or more than one co-morbidity, respectively.

**Table 1. Characteristics of patients at admission, by wave.**

| | | Wave 1 (n = 154) | Wave 2 (n = 409) | Total (n = 563) |
|---|---|---|---|---|
| **Sociodemographic** | | | | |
| **Sex** | Male | 86 (55.8) | 217 (53.1) | 303 (53.8) |
| | Female | 68 (44.1) | 192 (46.9) | 260 (46.2) |
| **Age, years** | <20 | 1 (0.7) | 4 (1.0) | 5 (0.9) |
| | 20–39 | 28 (18.2) | 65 (15.9) | 93 (16.5) |
| | 40–59 | 66 (42.9) | 156 (38.1) | 222 (39.4) |
| | 60–79 | 50 (32.5) | 147 (35.9) | 197 (35.0) |
| | ≥80 | 9 (5.8) | 37 (9.1) | 46 (8.2) |
| **Residence** | Harare | 147 (95.5) | 359 (87.8) | 506 (89.9) |
| | Outside Harare | 7 | 50 | 57 |
| **Comorbidities** | | | | |
| **HIV status (missing for 16)** | Negative | 136 (88.9) | 343 (87.1) | 479 (87.6) |
| | Positive | 17 (11.1) | 51 (12.9) | 68 (12.4) |
| **Diabetes** | No | 96 (62.8) | 289 (71.2) | 385 (68.9) |
| | Previously diagnosed | 56 (36.6) | 104 (25.6) | 160 (28.6) |
| | Newly diagnosed | 1 (0.7) | 13 (3.2) | 14 (2.5) |
| **Hypertension** | No | 67 (43.5) | 203 (50.0) | 270 (48.2) |
| | Previously diagnosed | 86 (55.8) | 203 (50.0) | 289 (51.6) |
| | Newly diagnosed | 1 (0.7) | 0 (0.0) | 1 (0.2) |
| **Tuberculosis (missing for 3)** | No | 150 (97.4) | 399 (98.3) | 549 (98.0) |
| | Previous | 3 (2.0) | 6 (1.5) | 9 (1.6) |
| | Current | 1 (0.7) | 1 (0.3) | 2 (0.4) |
| **Cardiovascular disease** | Yes | 22 (14.3) | 38 (9.4) | 60 (10.7) |
| **Asthma/COPD** | Yes | 5 (3.3) | 11 (2.7) | 16 (2.9) |
| **Chronic kidney disease** | Yes | 12 (7.8) | 31 (7.6) | 43 (7.7) |
| **Pregnancy** | Yes | 1 (0.7) | 16 (4.0) | 17 (3.1) |
| Comorbidity* | Yes | 116 (75.3) | 275 (67.2) | 391 (69.5) |
| **Oxygen saturation** | | | | |
| **On oxygen** | 88–100% | 18 (11.7) | 15 (3.7) | 33 (6.0) |
| | <88% | 13 (8.4) | 30 (7.3) | 43 (7.6) |
| **On room air** | <80% | 62 (40.3) | 144 (35.2) | 206 (36.6) |
| | 80-<88% | 25 (16.2) | 87 (21.3) | 112 (19.9) |
| | 88–100% | 36 (23.4) | 133 (32.5) | 169 (30.0) |
| **Weight category (missing for 28)** | Underweight | 9 (6.5) | 29 (7.3) | 38 (7.1) |
| | Normal | 92 (66.7) | 283 (71.3) | 375 (70.1) |
| | Overweight | 37 (26.8) | 85 (21.4) | 122 (22.8) |
| **Blood tests** | | | | |
| WBC count (10⁹/L) (missing for 29) | High (>11) | 52 (34.9) | 115 (29.9) | 167 (31.3) |
| **Hb (grams/decilitre) (missing for 28)** | Low (<13 if male, <12 if female, <11 if pregnant) | 55(36.7) | 145 (37.7) | 200 (37.4) |
| **Creatinine (micromol/litre) (missing for 45)** | High (>65.4 if male, >52.2 if female) | 13 (9.0) | 69 (18.5) | 82 (15.8) |
| | Normal | 72 (49.7) | 178 (47.7) | 250 (48.3) |

Abbreviations: HIV, Human Immunodeficiency Virus; TB, Tuberculosis; COPD, Chronic Obstructive Pulmonary Disease, WBC, white blood cell, Hb, Haemoglobin, HbA1c, glycosylated haemoglobin; ALT, alanine transaminase.

*Comorbidity: any of the following: cardiovascular disease, chronic kidney disease, diabetes, hypertension, HIV

**Table 2. Interventions and complications of COVID-19 patients, by outcome.**

|  |  | Discharged (N = 340) | Died (N = 214) | Total (n = 554) |
|---|---|---|---|---|
| **Complications** | Any complication | 90 (26.5) | 125 (58.4) | 215 (38.8) |
|  | Acute kidney injury | 32 (9.4) | 67 (31.3) | 99 (17.9) |
|  | Hyperglycaemia | 33 (9.7) | 39 (18.2) | 72 (13.0) |
|  | Venous thromboembolism | 22 (6.5) | 24 (11.2) | 46 (8.3) |
|  | Stroke | 5 (1.5) | 11 (5.1) | 16 (2.9) |
|  | Diabetic ketoacidosis | 5 (1.5) | 5 (2.3) | 10 (1.8) |
|  | Acute confusional state | 2 (0.6) | 4 (1.9) | 6 (1.1) |
|  | Multi-organ failure | 0 | 3 (1.4) | 3 (0.5) |
|  | Myocardial infarction | 2 (0.6) | 1 (0.5) | 3 (0.5) |
|  | End stage renal failure | 0 | 3 (1.4) | 3 (0.5) |
| **Interventions** | Any antibiotic | 124 (36.5) | 102 (47.7) | 226 (40.8) |
|  | Ceftriaxone | 79 (23.2) | 81 (37.9) | 160 (28.9) |
|  | Azithromycin | 60 (17.7) | 25 (11.7) | 85 (15.3) |
|  | Metronidazole | 17 (5.0) | 20 (9.4) | 37 (6.7) |
|  | Amoxicillin | 10 (2.9) | 8 (3.7) | 18 (3.3) |
|  | Dexamethasone | 227 (66.8) | 174 (81.3) | 401 (72.4) |
|  | Insulin | 36 (10.6) | 29 (13.9) | 65 (11.8‘) |
|  | HFNO (N = 48) or CPAP (N = 3)[a] | 12 (3.5) | 38 (17.8) | 50 (9.0) |
|  | Physiotherapy | 18 (5.3) | 10 (4.7) | 28 (5.1) |
|  | ICU | 5 (1.5) | 20 (9.4) | 25 (4.5) |
|  | Dialysis | 2 (0.6) | 7 (3.3) | 9 (1.6) |
|  | Intubation | 3 (0.9) | 4 (1.9) | 7 (1.3) |
|  | Blood transfusion | 2 (0.6) | 1 (0.5) | 3 (0.5) |

Abbreviations: HFNO, high-flow nasal oxygen; CPAP, continuous positive airway pressure; ICU, intensive care unit.

[a] one patient was trialled on CPAP before being swapped to HFNO so is in both categories.

Table 3 shows the results of univariate and multivariate negative binomial regression models of factors associated with mortality among participants who remained in the ward for more than one night. In univariate analysis, mortality was associated with older age, male sex, low oxygen saturation, being underweight, anaemia, high white blood count (WBC) count, high creatinine, one or more comorbidities (hypertension, diabetes, HIV, cardiovascular disease, chronic kidney disease) and not being pregnant. In multivariate analysis, older age, lower

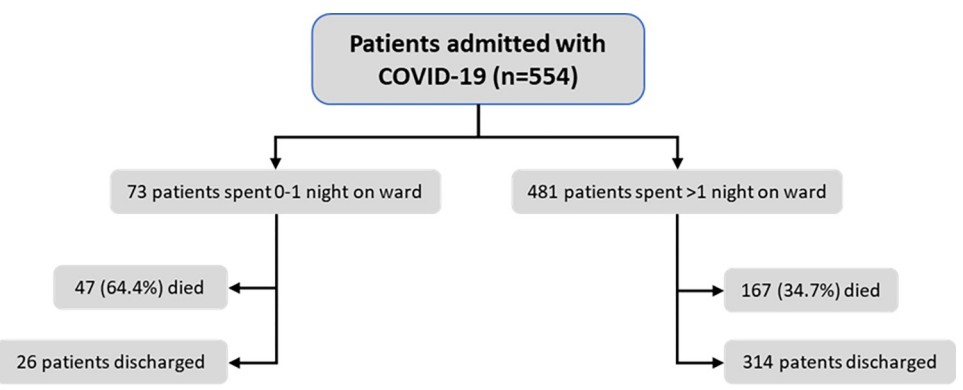

**Fig 2. Flowchart for admissions of COVID-19 patients.**

**Table 3. Factors associated with mortality in participants who stayed in the ward for more than 1 night, using univariate and multivariate negative binomial regression models.**

| Variable | Category | Died | Univariate (N = 481) | | | Multivariable (N = 426) | | |
|---|---|---|---|---|---|---|---|---|
| | | n/N (%) | RR (95% CI) | p-value | Wald test p-value | RR (95% CI) | p-value | Wald test p-value |
| Age | <20 | 0/5 | - | | <0.001 | - | | <0.001 |
| | 20-39 | 13/79 (16.5) | 1 | | | 1 | | |
| | 40-59 | 60/190 (31.6) | 1.92 (1.12, 3.29) | 0.018 | | 1.50 (0.95, 2.38) | 0.082 | |
| | 60-79 | 72/168 (42.9) | 2.60 (1.54, 4.41) | <0.001 | | 1.93 (1.22, 3.08) | 0.005 | |
| | 80-100 | 22/39 (56.4) | 3.43 (1.94, 6.06) | <0.001 | | 1.93 (1.13, 3.29) | 0.017 | |
| Sex | Male | 101/265 (38.1) | 1.25 (0.97, 1.61) | 0.087 | | | | |
| | Female | 66/216 (30.6) | 1 | | | | | |
| Location | Harare | 155/432 (35.9) | 1.47 (0.88, 2.44) | 0.141 | | | | |
| | Outside Harare | 12/49 (24.5) | 1 | | | | | |
| Oxygen saturation | <88% on oxygen | 26/34 (76.5) | 3.74 (2.57, 5.45) | <0.001 | <0.001 | 3.01 (2.05, 4.43) | <0.001 | <0.001 |
| | 88-100% on oxygen | 17/25 (68.0) | 3.33 (2.18, 5.08) | <0.001 | | 2.45 (1.46, 4.10) | 0.001 | |
| | <80% air | 75/175 (42.9) | 2.10 (1.45, 3.03) | <0.001 | | 1.67 (1.15, 2.41) | 0.006 | |
| | 80-<88% air | 20/105 (19.1) | 0.93 (0.56, 1.56) | 0.789 | | 0.85 (0.52, 1.38) | 0.506 | |
| | 88-100% air | 29/142 (20.4) | 1 | | | 1 | | |
| Weight category | Underweight | 20/33 (60.6) | 1.82 (1.33, 2.50) | <0.001 | <0.001 | 1.45 (0.99, 2.12) | 0.053 | 0.116 |
| | Normal | 106/319 (33.2) | 1 | | | 1 | | |
| | Overweight | 36/106 (34.0) | 1.02 (0.75, 1.39) | 0.890 | | 0.95 (0.71, 1.27) | 0.724 | |
| Hb (grams/dL) | Low (<13 if male, <12 if female, <11 if pregnant) | 71/177 (40.1) | 1.28 (1.00, 1.64) | 0.052 | | | | |
| | Normal | 91/290 (31.4) | 1 | | | | | |
| WBC (10$^9$/L) | Normal | 94/329 (28.6) | 1 | | <0.001 | 1 | | |
| | High (>11) | 68/137 (49.6) | 1.74 (1.37, 2.21) | <0.001 | | 1.59 (1.26, 2.00) | <0.001 | |
| Creatinine (micromol/litre) | Normal | 75/293 (25.6) | 1 | | | 1 | | |
| | High (>65.4 if male, >52.2 if female) | 81/159 (50.9) | 1.99 (1.55, 2.55) | <0.001 | | 1.63 (1.28, 2.07) | <0.001 | |
| Any comorbidity | No | 37/145 (25.5) | 1 | 0.008 | | | | |
| | Yes | 130/336 (38.7) | 1.52 (1.11, 2.07) | | | | | |
| Pregnancy | No | 165/466 (35.4) | 1 | | | | | |
| | Yes | 2/15 (13.3) | 0.38 (0.10, 1.38) | 0.140 | | | | |

Abbreviations: RR, risk ratio; 95% CI, 95% confidence interval; Hb, haemoglobin; WBC, white blood cell.

oxygen saturation, underweight, high WBC count and high creatinine remained associated with mortality. Table 4 shows the Cox regression for mortality hazard ratio using the same covariates. The association of age category with mortality hazard was not statistically significant on a Wald test. Otherwise, results were comparable to the results of the negative binomial regression.

Oxygen saturation, comorbidity (at least one of hypertension, diabetes, CKD, CVD and HIV) and pregnancy were associated with both mortality and wave. The unadjusted risk ratio (RR) for mortality in the second wave compared to the first was 1.03 (95% CI 0.81, 1.30; p = 0.83). After adjusting for confounders identified by a DAG (sex, oxygen saturation, comorbidities, and pregnancy), the adjusted RR was 1.23 (95%CI 1.00, 1.51; p = 0.05), suggesting that mortality was higher in the second wave than the first after adjusting for difference in patient population.

After 1$^{st}$ November 2020, lack of resources was felt by the treating clinical team to have contributed to 57 (35.4%) of 161 deaths. Lacking resources were ICU capacity (n = 35), HFNO

**Table 4. Factors associated with mortality in participants who stayed in the ward for more than 1 night, using multivariate Cox regression models.**

| Variable | Category | Multivariable (N = 434) | | |
|---|---|---|---|---|
| | | HR (95% CI) | p-value | Wald test p-value |
| **Age** | <20 | - | | 0.133 |
| | 20–39 | 1 | | |
| | 40–59 | 1.37 (0.78, 2.41) | 0.28 | |
| | 60–79 | 1.74 (0.99, 3.05) | 0.054 | |
| | 80–100 | 1.99 (0.98, 4.02) | 0.056 | |
| **Oxygen saturation** | <88% on oxygen | 2.97 (1.78, 4.96) | <0.001 | <0.001 |
| | 88–100% on oxygen | 2.39 (1.14, 5.03) | 0.022 | |
| | <80% air | 1.21 (0.79, 1.85) | 0.38 | |
| | 80-<88% air | 0.74 (0.43, 1.28) | 0.29 | |
| | 88–100% air | 1 | | |
| **Weight category** | Underweight | 1.72 (1.02, 2.91) | 0.043 | 0.113 |
| | Normal | 1 | | |
| | Overweight | 0.99 (0.65, 1.50) | 0.97 | |
| WBC ($10^9$/L) | Normal | 1 | | |
| | High (>11) | 1.52 (1.10, 2.11) | 0.012 | |
| **Creatinine (micromol/litre)** | Normal | 1 | | |
| | High (>65.4 if male, >52.2 if female) | 1.61 (1.17, 2.23) | 0.004 | |

Abbreviations: HR, Hazard Ratio

machines (n = 17) and theatre or dialysis staff (n = 5). Any patients desaturating (SpO2<88%) on low-flow oxygen (15 Litres via non-rebreather mask) would be offered HFNO, however for 17 patients all 4 machines were in use and they subsequently passed away. All patients who desaturated (SpO2<88%) on HFNO would be escalated to ICU if there was capacity. ICU capacity was limited due to a lack of ICU trained staff (anaesthetists and Intensive Care Nurses). ICU had a maximum capacity of 3 patients. If a patient required an operation or dialysis but there were no staff available, and they subsequently passed away this was counted in lack of theatre or dialysis group. In January 2021, when the unit admitted the highest number of patients, 45% of deaths (41/92) were attributed to lack of resources (Fig 3).

## Discussion

This study reports the clinical characteristics and outcomes of patients with COVID-19 in a tertiary public-sector hospital in Harare, Zimbabwe. The median age of Zimbabwe's population is estimated to be 18 years [12]. In comparison, the median age of patients admitted with COVID-19 was 56 years indicating that, as reported in other settings, age is an important risk factor for severe COVID-19 [13]. In addition, the difference between the median age of patients admitted to the Unit compared to the general population may explain the low population COVID-19 mortality in some African countries where only a small proportion of the population is >40 years of age [14]. More than two-thirds of patients had at least one comorbidity in keeping with data from other studies that show that older age and comorbidities are associated with a higher risk of severe COVID-19 [12, 15].

More than a third of patients presented with severe hypoxia (oxygen saturations below 80%), indicating delayed presentation. Likely reasons for late presentation include closure of public transport during lockdowns, shortage of ambulances, user fees (although these were removed for COVID-19) and concerns regarding medical costs, lack of awareness regarding

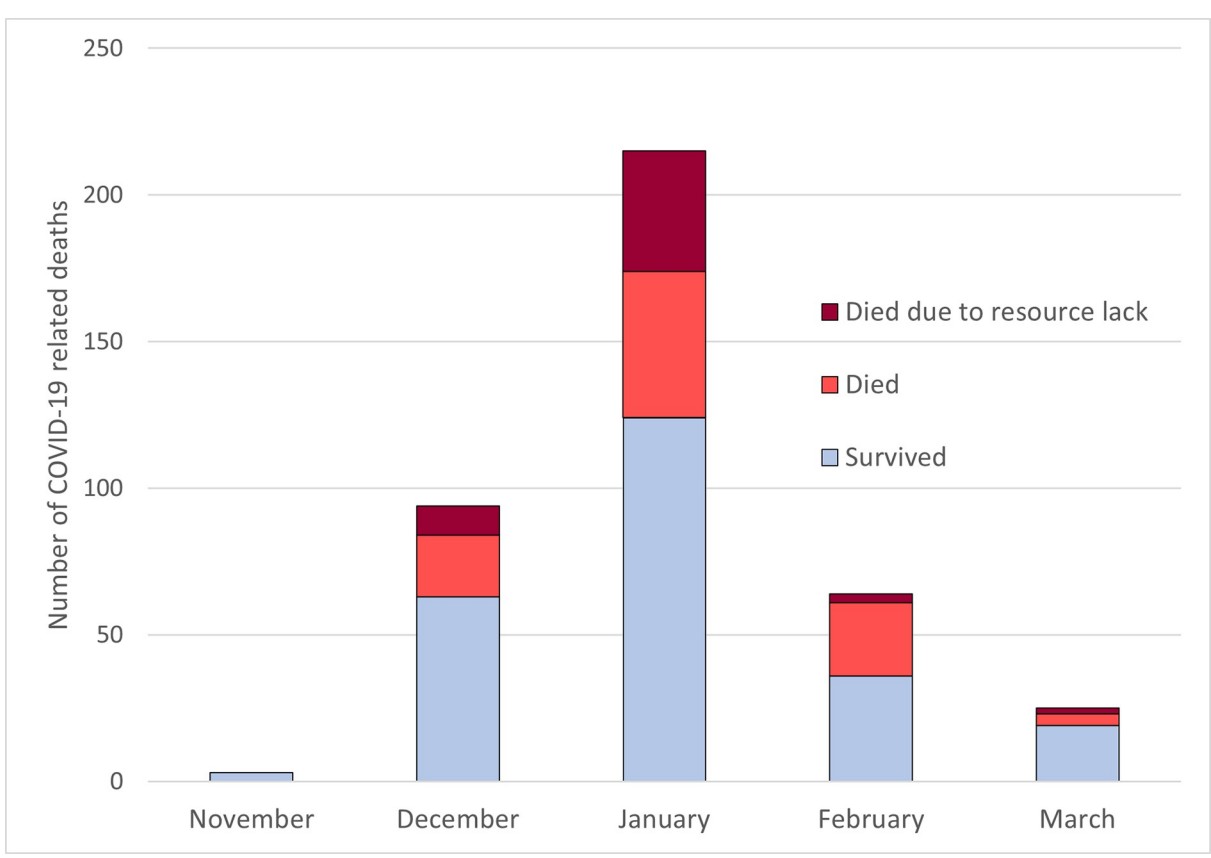

**Fig 3. Mortality attributable to lack of resources by month (November 2020 to March 2021, n = 409).**

the need for medical care and fear of coming to hospital during the pandemic [16, 17]. The late presentation may have contributed to the high mortality observed in this study.

Complications were common, occurring in nearly forty percent of patients. The most common were acute kidney injury, hyperglycaemia, and venous thromboembolism [18, 19]. Our study showed similar incidences of complications to studies performed in high and middle-income countries [20–22]. The high incidence of complications during admission emphasizes the importance of intensive monitoring of patients and management of comorbidities. Studies have shown that tight glycaemic control is associated with better outcomes in patients with COVID-19 and pre-existing diabetes [23–25].

Mortality among patients admitted with COVID-19 was 38%, which is comparable with data from other countries in the region [26–28]. Of note, 21% of deaths occurred within 24 hours of admission and may have been avoidable if patients had presented earlier. It is important to note that while evidence-based treatment protocols were implemented, care was delivered amid considerable health system challenges, and may have been uneven across patients, which may, in part, explain the high mortality.

Elevated white blood cell count and creatinine on admission, older age and being underweight were all independent risk factors for mortality, in line with other studies [29]. Notably, nearly a third of patients who died experienced an acute kidney injury compared to only 9.4% of those who were discharged. Acute kidney injury was strongly associated with mortality, consistent with other studies [13]. In this cohort, HIV was not associated with an increased risk of mortality. Zimbabwe has an adult HIV prevalence of 12.9% with coverage of antiretroviral therapy and viral suppression at 90.3% in the 2020 Population Based HIV Impact

Assessment [30]. In this cohort there were no new HIV diagnoses and all patients with HIV were on treatment.

During the first wave results of trials investigating effective treatments for patients with COVID-19 were published including steroids and prophylactic anticoagulation [9, 31, 32]. Studies in Europe and Asia showed either no change or a decrease in in-hospital mortality when comparing the second wave to the first attributed to better clinical management and improved knowledge on how to treat these patients [33–35]. In contrast, a South African study reported a higher in-hospital mortality in the second wave compared to the first explained by an older patient population, a new SARS-CoV-2 variant (Beta) and increased pressure on the health system [34]. Similar to the South African study, adjusted mortality in our study was higher in the second wave compared to the first wave. In the second wave, resources were stretched further and could have contributed to the higher observed mortality.

In the second wave, clinicians assessed over a third of deaths were attributable to lack of resources. It is important to note that the resources that were lacking were largely human resources rather than infrastructural. Zimbabwe has experienced an economic crisis over the past two decades with a hyperinflationary economy, which has had a substantial impact on the health system [36]. Even before the COVID-19 pandemic, there were recurrent strikes by healthcare workers and shortages in essential medicines with disruption of routine clinical services.

However, there were some key facilitators to providing care including the infrastructural planning that had occurred before the first wave, including bulk tanks for oxygen storage and installation of oxygen points within the unit, clear and simplified management protocols based on contemporary evidence that were contextually feasible, mentorship and supervision of the staff at the Unit and dedicated support from hospital management, including strengthening and supporting supply chains for essential medicines. Importantly, aspects of care that are sometimes neglected were prioritized, such as active palliative care for patients with end-stage disease and facilitating prompt burial arrangements, and engagement and communication with families (who were unable to visit their relatives) who provided integral feedback on how to improve quality of care. In our context, where care escalation options are limited, these aspects including active decisions about when to de-escalate care, are of even more importance.

We acknowledge several limitations. The study included data from one hospital only and results are not generalizable to more peripheral settings. However, at the time the study was conducted PGH was one of very few public sector hospitals providing in-patient COVID-19 care in the country. Given that some patients particularly during the second wave had to travel from far away, severity of COVID-19 was more extreme than experienced in countries with decentralized services. The study was conducted during the first two COVID-19 waves within a very busy and understaffed clinical service. Therefore, some data are missing and could not be retrospectively obtained from medical notes. Laboratory data was limited as patients and their families had to pay out of pocket for many blood tests. This may have resulted in under-ascertainment of conditions such as diabetes. The study assessed risk for mortality and complications based on clinical characteristics and did not include patient level analysis of access to advanced treatments (ICU, HFNO) which means outcomes may, in part, be explained by differences in clinical care.

In conclusion, successful management of COVID-19 needs to include evidence-based treatment, close monitoring, and prompt management of clinical complications. This in turn will require addressing health system constraints including health care worker shortages. Prevention of COVID-19 including vaccination, particularly among those with comorbidities, should be emphasized.

## Acknowledgments

The authors thank Dr Mapfanyangira and the Department of Anaesthetics at the Parirenyatwa Group of Hospitals and the Zimbabwe Ministry of Health and Child Care and Harare City Health Services for their support.

## Author Contributions

**Conceptualization:** Arun Fryatt, Katharina Kranzer, Rashida A. Ferrand.

**Data curation:** Arun Fryatt, Trevor Chivandire, Victoria Simms, Perseverance Chikide, Trymore Munorwa, Ellane Simon, Katharina Kranzer, Leolin Katsidzira, Rashida A. Ferrand.

**Formal analysis:** Victoria Simms, Lovemore Nyasha Sigwadhi, Rashida A. Ferrand.

**Funding acquisition:** Arun Fryatt, Rashida A. Ferrand.

**Investigation:** Arun Fryatt, Trevor Chivandire, Victoria Simms, Perseverance Chikide, Rashida A. Ferrand.

**Methodology:** Arun Fryatt, Trymore Munorwa, Ellane Simon, Rashida A. Ferrand.

**Project administration:** Arun Fryatt, Perseverance Chikide, Trymore Munorwa, Rashida A. Ferrand.

**Resources:** Arun Fryatt, Tsitsi M. Magure, Aspect Maunganidze, Leolin Katsidzira, Rashida A. Ferrand.

**Software:** Arun Fryatt, Victoria Simms.

**Supervision:** Arun Fryatt, Katharina Kranzer, Tsitsi M. Magure, Leolin Katsidzira, Rashida A. Ferrand.

**Validation:** Aspect Maunganidze, Rashida A. Ferrand.

**Visualization:** Arun Fryatt.

**Writing – original draft:** Arun Fryatt, Victoria Simms, Rashida A. Ferrand.

**Writing – review & editing:** Arun Fryatt, Trevor Chivandire, Victoria Simms, Perseverance Chikide, Trymore Munorwa, Ellane Simon, Lovemore Nyasha Sigwadhi, Katharina Kranzer, Tsitsi M. Magure, Aspect Maunganidze, Leolin Katsidzira, Rashida A. Ferrand.

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
