## [Decision Letter · Decision Letter 0]

10 May 2022

PGPH-D-22-00140

Clinical characteristics and outcomes of patients admitted with COVID-19 at a public-sector hospital over the first two waves of SARS-CoV2 infection in Harare, Zimbabwe

Dear Dr. Fryatt,

Thank you for submitting your manuscript to PLOS Global Public Health. After careful consideration, we feel that it has merit but does not fully meet PLOS Global Public Health’s publication criteria as it currently stands. Therefore, we invite you to submit a revised version of the manuscript that addresses the points raised during the review process.

Please submit your revised manuscript by . If you will need more time than this to complete your revisions, please reply to this message or contact the journal office at globalpubhealth@plos.org. Please include the following items when submitting your revised manuscript:

We look forward to receiving your revised manuscript.

Kind regards,

Abraham D. Flaxman, Ph.D.

Academic Editor

Journal Requirements:

1. Please provide an Author Summary. This should appear in your manuscript between the Abstract (if applicable) and the Introduction, and should be 150–200 words long. The aim should be to make your findings accessible to a wide audience that includes both scientists and non-scientists. Sample summaries can be found on our website under Submission Guidelines: 

https://journals.plos.org/globalpublichealth/s/submission-guidelines#loc-parts-of-a-submission

- State what role the funders took in the study. If the funders had no role in your study, please state: “The funders had no role in study design, data collection and analysis, decision to publish, or preparation of the manuscript.”

Additional Editor Comments (if provided):

Reviewers' comments:

Reviewer's Responses to Questions

**Comments to the Author**

1. Does this manuscript meet PLOS Global Public Health’s publication criteria? Is the manuscript technically sound, and do the data support the conclusions? The manuscript must describe methodologically and ethically rigorous research with conclusions that are appropriately drawn based on the data presented.

Reviewer #1: Yes

Reviewer #2: Yes

2. Has the statistical analysis been performed appropriately and rigorously?

Reviewer #1: Yes

Reviewer #2: Yes

3. Have the authors made all data underlying the findings in their manuscript fully available (please refer to the Data Availability Statement at the start of the manuscript PDF file)?

Reviewer #1: Yes

Reviewer #2: Yes

4. Is the manuscript presented in an intelligible fashion and written in standard English?

Reviewer #1: Yes

Reviewer #2: Yes

5. Review Comments to the Author

Reviewer #1: Thank you for the opportunity to review this manuscript by Fryatt et al. This well-written paper describes clinical characteristics of patients admitted with COVID-19 to a single urban public hospital in Harare, Zimbabwe, including complications and outcomes. The paper also touches on issues of hospital capacity and hypothesizes as to the impact of human resource shortages. Overall, this paper is informative, but the clinical characteristics are difficult to interpret with regard to their importance, and the discussion seems disconnected from the primary results of the manuscript. Below, please find additional detail.

Major Suggestions:

1. Risk factors associated with death and complications are reasonably performed. However, this study is complicated by the fact that clinical care seems to be uneven across patients. The authors talk about significant limitations in human resources, and as a result, limited advanced treatment (including NIPPV and intubation). However, this is not included in the patient level analysis. As a result, assessing risk for mortality and complications based on clinical characteristics alone is highly limited. and outcomes may actually be explained by differences in clinical care (at least in part). Additionally, the authors state in their conclusion that “evidence-based care was delivered,” yet later note challenges with supervision and limitations to escalate care. This is confusing and requires additional clarification.

2. The authors focus the majority of their discussion on the issue of limited human resources/capacity. However, this represents just 2 sentence in the results. Further, the way in which attribution of poor outcomes due to limited resources is based on a single provider’s limited documentation leaves many unanswered questions (the reason for lack of escalation is not shown here). I would recommend that the authors present the reasons for inability to escalate care and be clear about this cohort of patients, presenting additional information on their clinical characteristics that may have impacted outcome.

3. Were all patients admitted to the hospital tested for COVID at the time of admission? Please clarify. This will significantly impact the denominator. Additionally, in the results the authors note that 515 patients “reported” a test for COVID. Was this not abstracted from the chart? Who reported this?

4. Reframe the groups that are currently referred to as COVID and non-COVID to “admitted due to COVID” and “diagnosed with COVID, but admitted for an alternative reason” or some variation thereof.

5. 29 patients were PCR negative for COVID, yet included in this analysis. This needs to be further clarified in the methods.

6. The conclusion needs to be reframed to reflect the study results presented. I agree that the resource limitation issue is very interesting, but it is not the core study question and this study doesn’t adequately assess resource limitations as currently structured.

Minor suggestions:

1. Analyzing clinical characteristics is most useful in assessing risk of admission, complications or death. This paper only examines clinical characteristics of patients who are admitted, limiting any analysis to explain risk for hospitalization without a comparison to the general population. Doing this analysis would require additional investigation, but would be useful to front line clinicians in understanding risk for hospitalization and death (and hence need for higher level of care).

2. Characteristics of admitted patients (table 1). How was CVD identified? Patient reported history? How was CKD differentiated by AKI? I imagine that prior records were not available for many patients. I would suggest reframing these clinical characteristics to those present at admission rather than historical data. This will provide a better baseline for clinicians who will use this data to assess risk of decompensation among the patients admitted to their service.

3. Hyperglycemia as a complication is problematic. Agree that tight glucose control is important in critically ill patients, but not certain how useful this finding is in the context of medication induced hyperglycemia. Did all patients get dexamethasone?

4. Consider additional bins for WBC and Creatinine measurements. Normal vs. not normal is unlikely to be a clinically useful discriminatory tool.

5. The beginning of the fourth paragraph in the discussion section belongs in results, not discussion.

6. The fifth paragraph of the discussion, looking at the frequency of complications, would provide a nice opportunity to compare to other studies in both HIC and LMIC.

Reviewer #2: I would like to commend the authors for conducting this study. Studies on factors associated with COVID-19-related outcomes in low income countries are highly appreciated. I have some comments:

Major comments

1) Why not using Cox regression, instead of Poison regression, to asses hospital mortality?

2) Only 388 of 563 patients reported a positive PCR test/antigen test for SARS-COV-2. In this sense, please, describe, in methods section, the reasons and criteria used to classify participants with a negative or unknown/ inconclusive PCR/ antigen test result as COVID-19 patients.

Minor comments/suggestions

1) Please include the study design in the title;

2) Abstract: please include risk ratios with 95% CIs and p-values for the following comparison: "Mortality was higher in those with ≥1 comorbidity than in those without any (43.2% vs 28.2%)".

3)Methods - Study Design and setting: Please describe the number of hospital beds as well as the number of ICU beds (total and COVID-19 dedicated). Some information regarding structure/capacity to provide ventilatory support (eg, oxygen and mechanical ventilation) and health staff (eg patient/physician ratio and nurse/physician ratio) would facilitate the understanding of the local context.

4) Ethical approval: please describe the number of IRB approval;

5) Tables format: please variable definitions in footnotes (eg, weight category - table 1)

6) Table 1: I might suggest to include p-values for the comparisons between first and second waves;

7) Discussion - limitations: The authors listed the retrospective data collection as a limitation, but the study is prospective, please clarify. I would add the fact that some important variables that could be associated with outcomes were not measured (eg, corticosteroid use)

6. PLOS authors have the option to publish the peer review history of their article (what does this mean?). If published, this will include your full peer review and any attached files.

**Do you want your identity to be public for this peer review?** For information about this choice, including consent withdrawal, please see our Privacy Policy.

Reviewer #1: No

Reviewer #2: **Yes: **Regis Goulart Rosa

---

## [Decision Letter · Decision Letter 1]

19 Oct 2022

PGPH-D-22-00140R1

Clinical characteristics and outcomes of patients admitted with COVID-19 at a public-sector hospital over the first two waves of SARS-CoV2 infection in Harare, Zimbabwe: A prospective cohort study

Dear Dr. Fryatt,

Thank you for submitting your manuscript to PLOS Global Public Health. After careful consideration, we feel that it has merit but does not fully meet PLOS Global Public Health’s publication criteria as it currently stands. Therefore, we invite you to submit a revised version of the manuscript that addresses the points raised during the review process.

We look forward to receiving your revised manuscript.

Kind regards,

André Machado Siqueira, M.D., MSc, Ph.D

Academic Editor

Journal Requirements:

Additional Editor Comments (if provided):

The manuscript has improved considerably but there are still some comments made by reviewer #1 that deserve to be addressed.

Reviewers' comments:

Reviewer's Responses to Questions

**Comments to the Author**

1. If the authors have adequately addressed your comments raised in a previous round of review and you feel that this manuscript is now acceptable for publication, you may indicate that here to bypass the “Comments to the Author” section, enter your conflict of interest statement in the “Confidential to Editor” section, and submit your "Accept" recommendation.

Reviewer #2: All comments have been addressed

Reviewer #3: All comments have been addressed

2. Does this manuscript meet PLOS Global Public Health’s publication criteria? Is the manuscript technically sound, and do the data support the conclusions? The manuscript must describe methodologically and ethically rigorous research with conclusions that are appropriately drawn based on the data presented.

Reviewer #2: Partly

Reviewer #3: Yes

3. Has the statistical analysis been performed appropriately and rigorously?

Reviewer #2: No

Reviewer #3: Yes

4. Have the authors made all data underlying the findings in their manuscript fully available (please refer to the Data Availability Statement at the start of the manuscript PDF file)?

Reviewer #2: No

Reviewer #3: Yes

5. Is the manuscript presented in an intelligible fashion and written in standard English?

Reviewer #2: No

Reviewer #3: Yes

6. Review Comments to the Author

Reviewer #2: The authors addressed some reviewers comments on this revised manuscript version. Although the manuscript improved, some limitations still need to be addressed.

1) Since the authors use mortality as main outcome. I recommend to use a more accurate statistical test (eg, Cox regression) to verify consistency. It could be performed as main analysis or as a complementary sensitivity analysis.

2) There are some inconsistencies in manuscript writing. In this sense, I might suggest to edit the manuscript for grammar and style.

Reviewer #3: The authors have presented a real world example of healthcare provision to COVID-19 patients in Harare, Zimababwe during the first two waves of the COVID-19 pandemic in their country.

They are to be congratulated for doing this under trying circumstances.

The methodology is sound and descriptive in nature. The discussion of the results is appropriate.

The clinical characteristics and outcomes are well thought through and clearly presented.

The discussion around the resource limitiations in which this service operated are key to be able to lobby for additional health care resources in the future.

Specifically the statement: "After 1st November 2020, lack of resources was felt to have contributed to 57 (35.4%) of 161 deaths."

I feel should be qualified to be "After 1st November 2020, lack of resources was felt by the treating clinical team to have contributed to 57 (35.4%) of 161 deaths."

This is an interesting statement. It reflects the attitude of the clinicians in the field at the time. If another HIC healthcare system was making this assessment they may have another number all together. The other side to this statement is that 64.6% of deaths were felt to be expected even if additional resources were available. But when only +/- 5% of cases could access a ventilator most clinicians would say not enough time / resource investment has been made to be able to accurately make that assessment for a patient. This varies across healthcare systems greatly.

If the authors wish to highlight this they could add an additional sentence or two to the discussion.

Rest of the suggestions are minor and stylistic. See attached comments in PDF. The paper reads well to me and as with all such studies the context is important and I feel adequately described given the challenges of healthcare delivery in Africa.

7. PLOS authors have the option to publish the peer review history of their article (what does this mean?). If published, this will include your full peer review and any attached files.

**Do you want your identity to be public for this peer review?** For information about this choice, including consent withdrawal, please see our Privacy Policy.

Reviewer #2: No

Reviewer #3: **Yes: **David Thomson

---

## [Editor Report · Decision Letter 2]

12 Jan 2023

PGPH-D-22-00140R2

Clinical characteristics and outcomes of patients admitted with COVID-19 at a public-sector hospital over the first two waves of SARS-CoV-2 infection in Harare, Zimbabwe: A prospective cohort study

Dear Dr. Fryatt,

Thank you for submitting your manuscript to PLOS Global Public Health. After careful consideration, we feel that it has merit but does not fully meet PLOS Global Public Health’s publication criteria as it currently stands. Therefore, we invite you to submit a revised version of the manuscript that addresses the points raised during the review process.

We look forward to receiving your revised manuscript.

Kind regards,

André Machado Siqueira, M.D., MSc, Ph.D

Academic Editor
---

## [Decision Letter · Decision Letter 3]

18 Sep 2023

PGPH-D-22-00140R3

Clinical characteristics and outcomes of patients admitted with COVID-19 at a public-sector hospital over the first two waves of SARS-CoV-2 infection in Harare, Zimbabwe: A prospective cohort study

Dear Dr. Fryatt,

Thank you for submitting your manuscript to PLOS Global Public Health. After careful consideration, we feel that it has merit but does not fully meet PLOS Global Public Health’s publication criteria as it currently stands. Therefore, we invite you to submit a revised version of the manuscript that addresses the points raised during the review process.

We look forward to receiving your revised manuscript.

Kind regards,

Edna N Bosire, Ph. D

Academic Editor

Journal Requirements:

Additional Editor Comments (if provided):

Dear Authors,

You have improved this manuscript based on reviewer's comments.

In addition to reviewer 4'd comments, a few issues that need to be corrected before we can formally accept this manuscript includes:

Abstract- replace data was to "data were"

This study looks to add..., replace with this study aimed to.... or the aim of this study was to....

Line 90-91, a sentence reads "The time between ...was defined as first wave. Add Zimbabwe after first wave as this definition is based on Zimbabwe context and may not apply in other contexts.

Authors should ensure consistency in font size. E.g., check table 4 heading.

Write ICU in full the first time you use it - line 134

Discussion: Check for typos and punctuations. e.g. paragraph 3, correct punctuations before a sentence that reads "likely reasons for late presentation."

also, add a punctuation on "of note, 21% of deaths...."

Overall, the manuscript has been improved with the latest revisions. Authors should consider the minor revisions suggested before we can formally accept the manuscript for publication.

Reviewers' comments:

Reviewer's Responses to Questions

**Comments to the Author**

1. If the authors have adequately addressed your comments raised in a previous round of review and you feel that this manuscript is now acceptable for publication, you may indicate that here to bypass the “Comments to the Author” section, enter your conflict of interest statement in the “Confidential to Editor” section, and submit your "Accept" recommendation.

Reviewer #4: All comments have been addressed

2. Does this manuscript meet PLOS Global Public Health’s publication criteria? Is the manuscript technically sound, and do the data support the conclusions? The manuscript must describe methodologically and ethically rigorous research with conclusions that are appropriately drawn based on the data presented.

Reviewer #4: Yes

3. Has the statistical analysis been performed appropriately and rigorously?

Reviewer #4: Yes

4. Have the authors made all data underlying the findings in their manuscript fully available (please refer to the Data Availability Statement at the start of the manuscript PDF file)?

Reviewer #4: Yes

5. Is the manuscript presented in an intelligible fashion and written in standard English?

Reviewer #4: Yes

6. Review Comments to the Author

Reviewer #4: Title of the manuscript: Clinical characteristics and outcomes of patients admitted with COVID-19 at a public sector hospital over the first two waves of SARS-CoV-2 infection in Harare, Zimbabwe: A prospective cohort study.

Number of the manuscript: PGPH-D-22-00140R3.

Reviewer: Christian B. Ngandu

Date: September 10, 2023.

Reviews comments

General comments

• The paper is well designed and very well written. There is a consistency between the study aim with sections of the manuscript. All variables need definitions.

• Keywords are too many. They should be reduced in number.

Specific comments

Abstract

• I would suggest the following sentence for the methods section (Line 37-38): “…mortality was higher in patients with comorbidity compared to those without any comorbidity…”

• For consistency with the title, I would suggest the following (Line 29); ‘…. the association between sociodemographic and clinical characteristics and mortality ….

Introduction

• The style of in-text citations brackets should be consistent. Please, edit on line 71 according to the whole text?

Methodology

• Regression methods is fitting with the type of data Variables; steps have been followed. However, none has been said, in the method section, about how the authors tried managing potential over-dispersion or missing of data. This should be said clearly.

• The manuscript needs clear definitions of variables. I suggest variable definitions for better understanding of the manuscript. For illustration, elsewhere nutrition status of participants should be defined and measures for assessment described. Also, dependent variable and independent variables should be clearly mentioned in the methods section.

• Line 140: I suggest the authors to insert this relevant citation for the Poisson regression with robust errors (Cameron, A. C. and Trivedi, P. K. (2009). Micro econometrics Using Stata. College Station, TX: Stata Press.)

Results

• Table 2:

o the total values lines ‘Any complications and ‘Dexamethasone’ need some editing. The authors can, please, check for and edit?

• Poisson regression in Table 3

o Category age under twenty years should be deleted in that sense this makes a bit confusion at reading the table 3 and table 4.

• Cox regression in Table 4:

o Category age under twenty years should be deleted in that sense this makes a bit confusion at reading the table 4.

o We have mentioned an Effect dose for oxygen saturation on mortality. This result is consistent and comfortable with other evidence.

o Underweighted individuals were more likely to die compared to overweight. I would like the measure of overweight being defined in the method section for better understating. Also, this contrasting result should better be discussed and provide clarity in the discuss section.

• Limitations of the study. It has been noticed that missing data were recorded in the study. How the authors managed to alleviate the negatives effects of missing data?

References

• The authors should check for the consistency in the reference list at using the capital letters for the title of the resources. See references numbers 8, 9, 14, 19, 23, 25, 28, 31.

7. PLOS authors have the option to publish the peer review history of their article (what does this mean?). If published, this will include your full peer review and any attached files.

**Do you want your identity to be public for this peer review?** For information about this choice, including consent withdrawal, please see our Privacy Policy.

Reviewer #4: No

---

## [Editor Report · Decision Letter 4]

22 Nov 2023

Clinical characteristics and outcomes of patients admitted with COVID-19 at a public-sector hospital over the first two waves of SARS-CoV-2 infection in Harare, Zimbabwe: A prospective cohort study

PGPH-D-22-00140R4

Dear Dr Fryatt,

We are pleased to inform you that your manuscript 'Clinical characteristics and outcomes of patients admitted with COVID-19 at a public-sector hospital over the first two waves of SARS-CoV-2 infection in Harare, Zimbabwe: A prospective cohort study' has been provisionally accepted for publication in PLOS Global Public Health.

Best regards,

Ting Shi

Academic Editor